# Tau monomer encodes strains

**Apurwa M Sharma[1,2], Talitha L Thomas[1], DaNae R Woodard[1], Omar M Kashmer[1], Marc I Diamond[1]\***

[1]Center for Alzheimer's and Neurodegenerative Diseases, University of Texas Southwestern Medical Center, Dallas, United States; [2]Graduate Program in Biochemistry, Division of Biology and Biomedical Sciences, Washington University in St Louis, St. Louis, United States

**Abstract** Tauopathies have diverse presentation, progression, and neuropathology. They are linked to tau prion strains, self-replicating assemblies of unique quaternary conformation, whose origin is unknown. Strains can be propagated indefinitely in cultured cells, and induce unique patterns of transmissible neuropathology upon inoculation into mice. DS9 and DS10 cell lines propagate different synthetic strains that derive from recombinant tau. We previously observed that tau monomer adopts two conformational states: one that is inert ($M_i$) and one that is seed-competent ($M_s$) (*Mirbaha et al., 2018*). We have now found that $M_s$ itself is comprised of multiple stable ensembles that encode unique strains. DS9 monomer inoculated into naive cells encoded only DS9, whereas DS10 monomer encoded multiple sub-strains. Sub-strains each induced distinct pathology upon inoculation into a tauopathy mouse model (PS19). $M_s$ purified from an Alzheimer's disease brain encoded a single strain. Conversely, $M_s$ from a corticobasal degeneration brain encoded three sub-strains, in which monomer from any one re-established all three upon inoculation into cells. Seed competent tau monomer thus adopts multiple, stable seed-competent conformations, each of which encodes a limited number of strains. This provides insight into the emergence of distinct tauopathies, and may improve diagnosis and therapy.
DOI: https://doi.org/10.7554/eLife.37813.001

**\*For correspondence:**
Marc.Diamond@UTSouthwestern.edu

**Competing interests:** The authors declare that no competing interests exist.

## Introduction

Tauopathies are a diverse group of neurodegenerative diseases defined by the accumulation of tau amyloids in neurons and glia (*Lee et al., 2001*). They include Alzheimer's disease (AD) and cortico-basal degeneration (CBD), among many clinical and neuropathological syndromes. Most are sporadic, and some are caused by dominant mutations in the microtubule associated protein tau (MAPT) gene (*Lee et al., 2001*). The origin of different sporadic tauopathies is poorly understood. We initially hypothesized that they might arise from distinct, self-propagating assemblies based on our observation that a single tau monomer stably propagates different structures in vitro, depending on the template to which it is initially exposed (*Frost et al., 2009a*). Experiments from our group and others subsequently indicated that tau transmits amyloid pathology into cultured cells and mouse brain that can move from cell to cell (*Frost et al., 2009b*; *Clavaguera et al., 2009*), and led to the idea that it is 'prion-like.'

Prion protein (PrP) prions form 'strains,' which are distinct amyloid structures derived from a single protein that propagate indefinitely in living systems, and which underlie diverse and predictable patterns of neuropathology in humans and mice. We found that tau forms strains that can be isolated and propagated from cell to cell in a stable line that expresses the repeat domain (RD) containing two disease-associated mutations (P301L/V337M) fused to yellow fluorescent protein (RD-YFP). We initially created two 'artificial' strains based on inoculation of recombinant fibrils into RD-YFP cells and isolation of two clonal lines that stably propagated aggregates of distinct conformation: DS9 and DS10. Inoculation of DS9 and DS10 lysates into a mouse model of tauopathy

(PS19) that is based on expression of 1N4R human tau with a single disease-associated mutation (P301S) (*Yoshiyama et al., 2007*), created distinct neuropathological phenotypes. These could be transmitted stably across multiple generations of mice, and finally back into the RD-YFP biosensor cells, where DS9 and DS10 strains displayed their original morphology (*Sanders et al., 2014*). Additionally, using the RD-YFP biosensor cells we isolated distinct tau strain ensembles from the brains of patients with different tauopathies. This linked different strains to the various neuropathological syndromes. Unlike the prion protein (PrP), tau is not known to cause infectious neurodegenerative disease in humans. However, after creating two classes of infectious proteinaceous particles from recombinant protein that accounted for distinct and stably transmissible patterns of neuropathology, we proposed that tau should be considered a prion (*Sanders et al., 2014*).

Subsequently, we created 18 distinct tau prion strains and observed that following intracerebral inoculation in mice they created distinct pathological patterns and rates of progression (*Kaufman et al., 2016*). Indeed, others had previously made similar observations of neuropathological diversity upon inoculation of tau fibril preparations from human tauopathies (*Clavaguera et al., 2013*). We have concluded that, in addition to probable contributions of environmental and genetic factors, the diversity of tauopathy can be explained in large part by the strains that underlie them, in other words, by protein conformation (*Kaufman et al., 2016*).

In work to define the minimal infectious unit, or 'seed' of tau prions, we previously determined that tau monomer adopts multiple, stable structures that we have grouped into two general categories: 'inert' ($M_i$) or 'seed-competent' ($M_s$) (*Mirbaha et al., 2018*). Although $M_i$ does not spontaneously self-assemble or seed aggregation, $M_s$ has a distinct conformation that allows self-assembly and seeding activity. We have isolated $M_s$ from recombinant sources and human brain. It retains its activity following isolation by size exclusion chromatography (SEC) and passage through a 100kD cutoff filter (*Mirbaha et al., 2018*). Structural studies (*Mirbaha et al., 2018*) suggest that $M_s$ is differentiated from $M_i$ based on exposure of critical amino acids ($^{275}$VQIINK$^{280}$/$^{306}$VQIVYK$^{311}$) that have previously been determined to mediate amyloid formation (*von Bergen et al., 2000*; *von Bergen et al., 2001*). In $M_i$ they are predicted to be buried in hairpins and rendered relatively inaccessible for intermolecular interactions.

The existence of $M_s$ as a unique conformational ensemble raised the question of what is the role of monomer in strain formation. We considered two possibilities. First, a single structure of $M_s$ might underlie multiple distinct assemblies. This has been suggested as the basis of two different fibril morphologies isolated from an AD patient, in which the same essential monomeric building block appears to constitute the 'core' of the amyloid in both structures (*Fitzpatrick et al., 2017*). Second, an $M_s$ protein might adopt multiple distinct conformations as an ensemble, each producing a single strain or subset of strains upon self-association in a multimeric assembly. This would predict that the diversity of tau prion strains we have previously described (*Sanders et al., 2014*; *Kaufman et al., 2016*) could be linked back to a distinct set of conformers of tau monomer. We have addressed this question by studying strains propagated in HEK293T cells, DS9 and DS10, that were derived from recombinant fibrils, and others derived from AD and CBD brain.

## Results

### Tau monomer dictates strain identity

We previously characterized in detail two tau strains, DS9 and DS10 (*Sanders et al., 2014*). We hypothesized that if $M_s$ acts as an 'unrestricted' building block, monomer derived from either DS9 or DS10 would produce a diversity of strains. We isolated total lysate or monomer from DS9 by using size exclusion chromatography (SEC) or by passage through a 100kD size-exclusion filter, methods that we have previously determined to faithfully exclude larger assemblies (*Mirbaha et al., 2018*). We then inoculated DS1 cells, which lack any aggregates, and used FACS to isolate single aggregate-containing cells, with which we established monoclonal lines for further analysis. We initially used epifluorescence microscopy to characterize the various colonies for inclusion morphology, which serves as a rough surrogate for strain identity (*Sanders et al., 2014*; *Kaufman et al., 2016*). For $M_s$ derived from DS9, we observed no variation—each of 52 clones exhibited the speckled

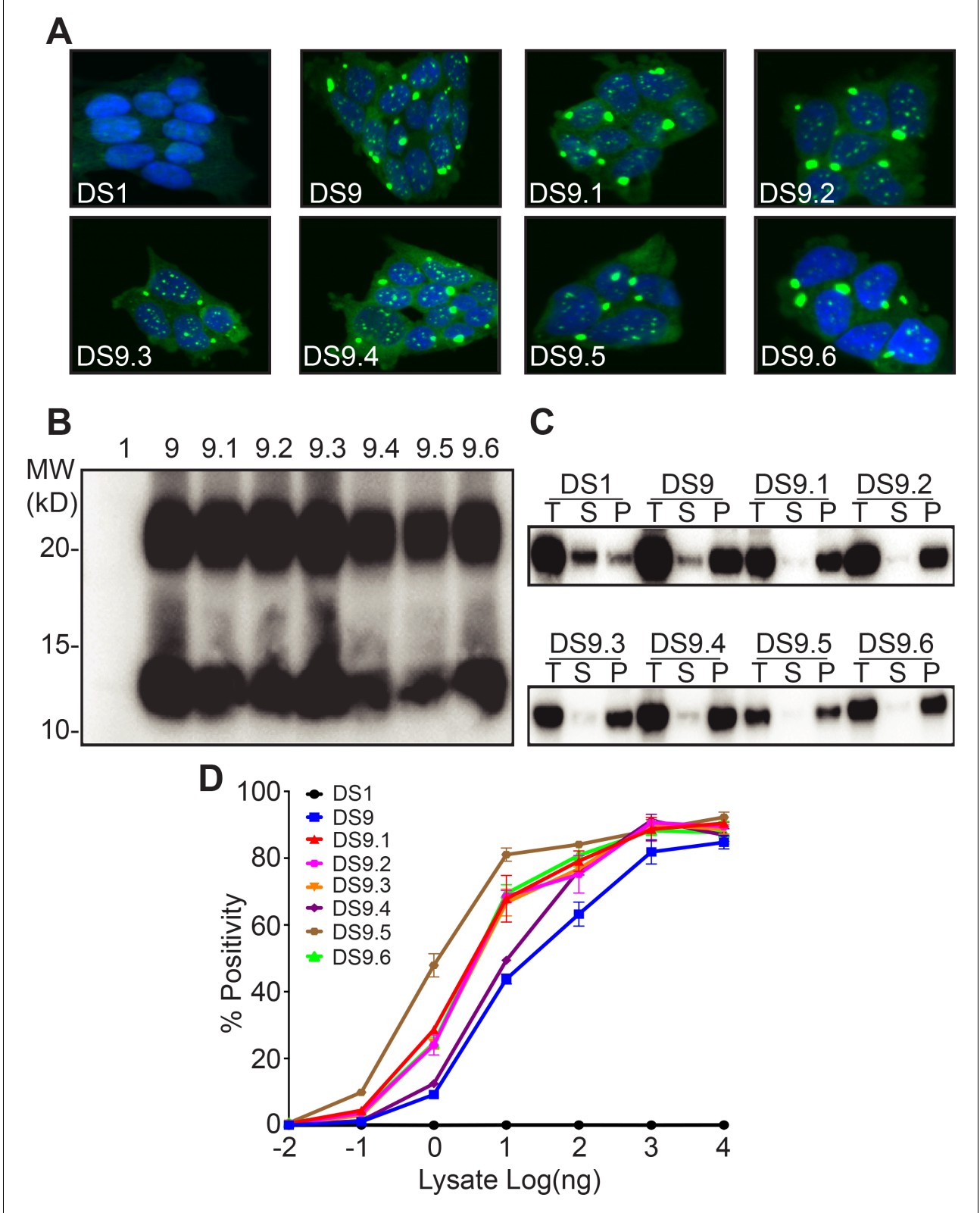

**Figure 1.** Tau M$_s$ from DS9 retains strain identity. (**A**) Clones isolated from DS9 monomer (9.1–9.6) show morphological characteristics similar to DS9. (**B**) Limited proteolysis digests all the monomer from DS1, but reveals similar protease resistant band patterns for DS9 and DS9.1–9.6. Both DS9 and its sub-strains exhibited a band around 10 kD, and a second band between 20 and 25 kD. (**C**) Sedimentation analysis was performed on DS1, 9, and its substrains DS9.1–9.6. Total (T) lysate was resolved into supernatant (S) and pellet (P) fractions by ultracentrifugation. Supernatant to pellet ratio

*Figure 1 continued on next page*

Figure 1 continued

loaded on the gel was 1:1 for all samples. DS1 had tau in the supernatant, whereas DS9 and its substrains had tau predominantly in the pellet. The band represents RD-YFP at ~45 kD. (D) DS9 and sub-strains had similar seeding activities upon transduction into P301S FRET biosensor cells. Images are representative of thousands similar cells. Western blots are representative of at least three replicates. Seeding assays represent an individual experiment in which each data point represents a sample analyzed in triplicate. Error bars represent the standard deviation.

DOI: https://doi.org/10.7554/eLife.37813.002

The following source data is available for figure 1:

**Source data 1.** Source data for *Figure 1D*.

DOI: https://doi.org/10.7554/eLife.37813.003

conformation previously observed for DS9 (*Sanders et al., 2014*) (*Figure 1A*). We isolated six typical 'sub-strains,' termed DS9.1–9.6, for further characterization in detail. In addition, we passed the lysate through a 100 kD size exclusion filter and used it to inoculate DS1 cells, creating 31 monoclonal lines. All were indistinguishable from DS9 by confocal microscopy. Inoculation using unfractionated DS9 lysate, or lysate of any of the sub-strains produced a single population of clones, all identical to DS9 in morphology (*Figure 1A*).

We have previously used proteolysis of insoluble tau to discriminate distinct strains. This reveals variation in the protease resistant 'cores' of tau aggregates (*Sanders et al., 2014*; *Kaufman et al., 2016*). DS1 had no protease resistant species. DS9 and DS9.1–9.6 exhibited very similar limited proteolysis patterns, with bands at 10 kD, and between 20 and 25 kD (*Figure 1B*). Next we used sedimentation analysis to differentiate the clones by subjecting the clarified lysate to high-speed centrifugation to separate the soluble from insoluble species. DS9 and DS9.1–9.6 exhibited similar fractionation patterns, with most tau being insoluble (*Figure 1C*). Finally, we used an established biosensor cell line that expresses tau RD (P301S) fused to cyan and yellow fluorescent proteins (RD-CFP/YFP) to monitor the ability of strains to trigger intracellular aggregation. DS9 and DS9.1–9.6 had identical maximal seeding, with variation in the dose responses within ~1 log of concentration, which is typical for independent isolates of the same strain (*Figure 1D*). Thus, monomer from DS9 faithfully encoded six identical DS9 sub-strains.

## DS10 monomer encodes multiple sub-strains

Extending our studies with DS9, we isolated total lysate or $M_s$ from DS10, transduced DS1 cells, and isolated multiple monoclonal inclusion-bearing lines. As we have previously reported, inoculation with unfractionated DS10 lysate produced a single population of clones, all identical to DS10 (*Sanders et al., 2014*). However, DS10 monomer created five distinct sub-strains, easily discerned by inclusion morphology: 36% were ordered (and indistinguishable from the parent strain) termed DS10.1; 21% were speckled, termed DS10.2; 13% were thread-like, termed DS10.3; 9% were disorganized, termed DS10.4 (*Figure 2A*, *Table 1*). Approximately 21% of cells formed a fifth strain, DS10.5, that rapidly 'sectored,' that is, lost its inclusions, and thus could not be further characterized. All strain images were analyzed by a blinded reviewer (D.R.W.). DS10 monomer was also isolated using a 100 kD cutoff filter. After transduction into DS1 cells we isolated monoclonal lines, and obtained similar strain diversity (*Table 2*).

We used limited proteolysis to compare the four monoclonal DS10 sub-strains (*Figure 2B*). DS10 and DS10.1 exhibited similar protease resistant doublets around 10 kD and a relatively light band around 20 kD. DS10.2 exhibited bands at 10 kD and between 20 and 25 kD. DS10.3 and 10.4 exhibited a band at 15 kD. DS10.3 exhibited a band at 10 kD, which was mostly absent in DS10.4. Sedimentation analysis helped further discriminate the sub-strains: DS10, DS10.1 and DS10.2 primarily contained an insoluble fraction and a small soluble fraction. By contrast, DS10.3 and DS10.4 exhibited mixed solubility, with about half soluble and insoluble tau (*Figure 2C*). DS10 sub-strains also had very different seeding efficiency that spanned >2 log orders of concentration (*Figure 2D*). DS10 and DS10.1 had almost identical seeding profiles, with an $EC_{50}$ of 300 ng for clarified lysate. DS10.2 had an $EC_{50}$ of 10 ng. DS10.3 had a much weaker seeding activity with an $EC_{50}$ at >10 µg (unable to be determined accurately), and reduced maximal seeding efficiency at ~50%. DS10.4 had an $EC_{50}$ of 100 ng. Based on multiple measures, we concluded that unlike DS9, DS10 monomer encoded distinct sub-strains, each with unique morphological and biochemical characteristics.

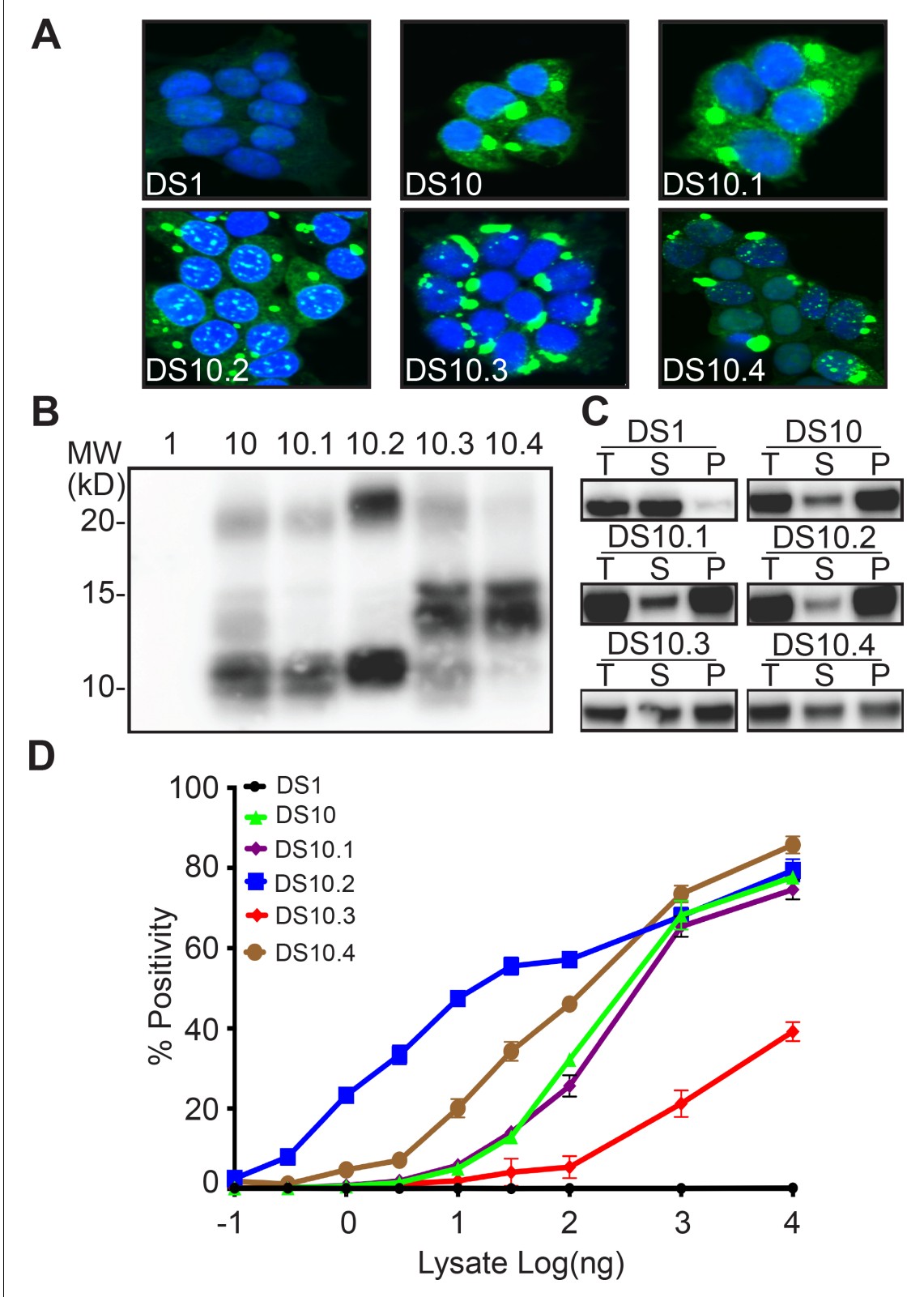

**Figure 2.** Tau M$_s$ from DS10 creates multiple sub-strains. (**A**) Clones isolated from DS10 monomer give rise to cells with multiple morphologies. Four sub-strains were discriminated based on multiple tests. (**B**) Limited proteolysis of RD-YFP using pronase differentiated the protease resistant cores in the sub-strains. Lane 1 represents DS1, which is comprised of RD-YFP monomer that is completely digested. (**C**) Sedimentation analysis of RD-YFP was performed on DS1, 10, and DS10.1–4. Total (T) lysate was resolved into supernatant (S) and pellet (P) fractions by ultracentrifugation. Supernatant to

*Figure 2 continued*

pellet ratio loaded on the gel was 1:1 for all samples. DS1 had RD-YFP in the supernatant; DS10, 10.1 and 10.2 had most RD-YFP in the pellet; DS10.3 and DS10.4 had mixed RD-YFP solubility. (D) DS10 sub-strains had distinct seeding activities upon transduction into P301S FRET biosensor cells. Images are representative of thousands similar cells. Western blots are representative of at least three replicates. Seeding assays represent an individual experiment in which each data point represents a sample analyzed in triplicate. Error bars represent the standard deviation.

DOI: https://doi.org/10.7554/eLife.37813.004

The following source data is available for figure 2:

**Source data 1.** Source data for *Figure 2D*.

DOI: https://doi.org/10.7554/eLife.37813.005

## Conformational ensembles of $M_s$

Sub-strains of DS10 could arise because multiple distinct monomers co-exist, each constrained to form only a single sub-strain. In this case, $M_s$ from each sub-strain would be predicted to recreate only that sub-strain (as for DS9.1–9.6). Alternatively, $M_s$ derived from DS10 might exist as a conformationally restricted ensemble, with relatively low kinetic and energetic barriers between various states that become 'locked in' upon higher-order assembly. In this case, monomer from any sub-strain would establish all of the others upon re-inoculation. To test these ideas, we isolated monomer from each of the DS10 sub-strains, re-introduced them into DS1, and isolated n = 47–51 monoclonal cell lines bearing inclusions from each inoculation. We analyzed each of these using blinded scoring of intracellular inclusion morphology. In 49 sub-strains derived from DS10.1 monomer, 45 appeared identical to DS10.1, except for four that formed DS10.5 (sectored) (*Table 3*). This suggested $M_s$ from DS10.1 was relatively restricted in its conformation. We obtained very different results with the other DS10 sub-strains. $M_s$ derived from each of DS10.2, DS10.3, or DS10.4 gave rise to all sub-strains, including DS10.1, with predomination of the original sub-strain (*Table 3*). This indicated that $M_s$ from these DS10 substrains existed as conformational ensemble. Taken together, our experiments revealed that while multimeric assemblies from total lysate exhibit consistent strain behavior in all cases, that is, faithful replication, tau monomer ($M_s$) is less constrained. While $M_s$ derived from DS9 was completely restricted to forming a single strain, and $M_s$ from DS10.1 predominantly encoded one sub-strain, others (DS10.2, DS10.3, DS10.4) adopted a defined set of strains in which $M_s$ from any could give rise to all five.

## Sub-strains induce distinct pathologies in PS19 mice

*Bona fide* tau prion strains create distinct and predictable pathologies *in vivo*. We previously used inoculation into PS19 mice that express full-length (1N,4R) human tau with a P301S mutation (*Yoshiyama et al., 2007*) to study the pathology of strains (*Sanders et al., 2014*; *Kaufman et al., 2016*). To test the activities of sub-strains, we inoculated n = 5 replicates of DS9, DS9.1, DS10, DS10.1–10.4 into the left hippocampus of PS19 mice at 3mos. Mice were age- and gender-matched to the extent possible, and derived from multiple independent litters. 4 weeks after inoculation, we analyzed the brains of the mice by staining with AT8 antibody, which recognizes p-Ser202 and p-Thr205 of tau (*Figure 3*).

Mice inoculated with DS1 controls exhibited no tau pathology in the hippocampus, CA1, or CA3. DS9, DS10, and all sub-strains induced different types of pathology. DS9 and DS9.1 induced tangle-

**Table 1.** Sub-strains generated from DS9 monomer isolated by SEC or cutoff filter.

Tau RD-YFP monomer ($M_s$) was isolated from DS9 either by SEC or 100kD cutoff filter and inoculated into DS1 to create sub-strains. Multiple clones were isolated and characterized by morphology. Columns indicate the number of clones identified (n) and the percentage this represents of the total (%). A single sub-strain was observed regardless of purification method. Classification of cell morphology was performed using blinded analysis.

| $M_s$ | SEC | | 100kD filter | |
| --- | --- | --- | --- | --- |
| | N | % | N | % |
| 9.1 | 52 | 100 | 31 | 100 |

DOI: https://doi.org/10.7554/eLife.37813.006

**Table 2.** Sub-strains generated from DS10 monomer isolated by SEC or cutoff filter.

Tau RD-YFP monomer ($M_s$) was isolated from DS10 by either by SEC or 100kD cutoff filter and inoculated into DS1 to create sub-strains. Multiple clones were isolated and characterized by morphology. Columns indicate the number of clones identified (n) and the percentage this represents of the total (%). Isolation of $M_s$ from DS10 by SEC or 100 kD cutoff filter each enabled a similar proportion of sub-strains to form. Classification of cell morphology was performed using blinded analysis.

| $M_s$ | SEC | | 100kD filter | |
|---|---|---|---|---|
| | N | % | N | % |
| 10.1 | 19 | 36 | 12 | 43 |
| 10.2 | 11 | 21 | 3 | 11 |
| 10.3 | 7 | 13 | 5 | 20 |
| 10.4 | 5 | 9 | 3 | 11 |
| 10.5 (sectored) | 11 | 21 | 4 | 15 |
| Total | 53 | 100 | 27 | 100 |

DOI: https://doi.org/10.7554/eLife.37813.007

like inclusions throughout CA1 and CA3 in patterns that were indistinguishable from one another, consistent with prior observations (*Sanders et al., 2014*). As expected, DS10 induced AT8-positive puncta in mossy fiber tracts of the hippocampus with strong staining patterns in the cell body, and AT8-positive puncta in CA3. DS10.1 induced pathology similar to DS10 in both CA1 and CA3. DS10.2 induced tangle-like inclusions throughout CA1, but with little or no pathology in CA3. DS10.3 had very low seeding efficiency and induced pathology only in CA1. DS10.4 also had low seeding and induced pathology only in CA1. A blinded reviewer (O.M.K.) analyzed slides derived from all inoculations (n = 40), and classified them by group. DS9 and DS9.1 were indistinguishable. DS10.1–10.3 were all easily distinguished, with a single error for DS10.4, which was misclassified in one instance (*Table 4*).

## Distinct $M_s$ conformations in AD and CBD brains

The preceding studies with DS9 and DS10 were based on tau RD-YFP fusions, and left uncertain whether tau monomer derived from human tauopathies would similarly encode strains. We have previously determined strain composition in AD and CBD patient brains to be quite distinct (*Sanders et al., 2014*). Thus we used a representative brain from each as a source of tau monomer. We gently lysed the brains by dounce homogenization, a method previously determined not to liberate $M_s$ from pre-existing fibrils (*Mirbaha et al., 2018*), and used an anti-tau antibody directed at the N-terminus (HJ8.5) to purify full-length tau and resolve monomer by SEC. We inoculated DS1 cells with total lysate from an AD patient and recovered a single clonal morphology, AD(t). We also used AD $M_s$ isolated by SEC to inoculate DS1 cells and recovered an identical clonal morphology, AD(m).

**Table 3.** Quantification of second generation of sub-strains obtained from DS10.

$M_s$ from each sub-strain of DS10 (10.1–10.5) was inoculated into DS1, and clones of the induced strains were characterized. DS10.1 largely produced a single predominant strain identical to DS10.1 (92%) and another strain DS10.5 that rapidly sectored (8%). DS10.2–10.4 each recreated all other strains. Columns indicate the number (n) of clones characterized and the percentage of the total (%) in each case. Classification of cell morphology was performed using blinded analysis.

| | Induced clone | | | | | | | | | | | |
|---|---|---|---|---|---|---|---|---|---|---|---|---|
| | 10.1 | | 10.2 | | 10.3 | | 10.4 | | 10.5 (sectored) | | Total | |
| $M_s$ | n | % | n | % | n | % | n | % | n | % | n | % |
| 10.1 | 45 | 92 | 0 | 0 | 0 | 0 | 0 | 0 | 4 | 8 | 49 | 100 |
| 10.2 | 3 | 6 | 37 | 79 | 2 | 4 | 2 | 4 | 3 | 7 | 47 | 100 |
| 10.3 | 11 | 21 | 3 | 6 | 21 | 42 | 4 | 8 | 12 | 23 | 51 | 100 |
| 10.4 | 17 | 35 | 3 | 7 | 13 | 27 | 12 | 24 | 3 | 7 | 48 | 100 |

DOI: https://doi.org/10.7554/eLife.37813.008

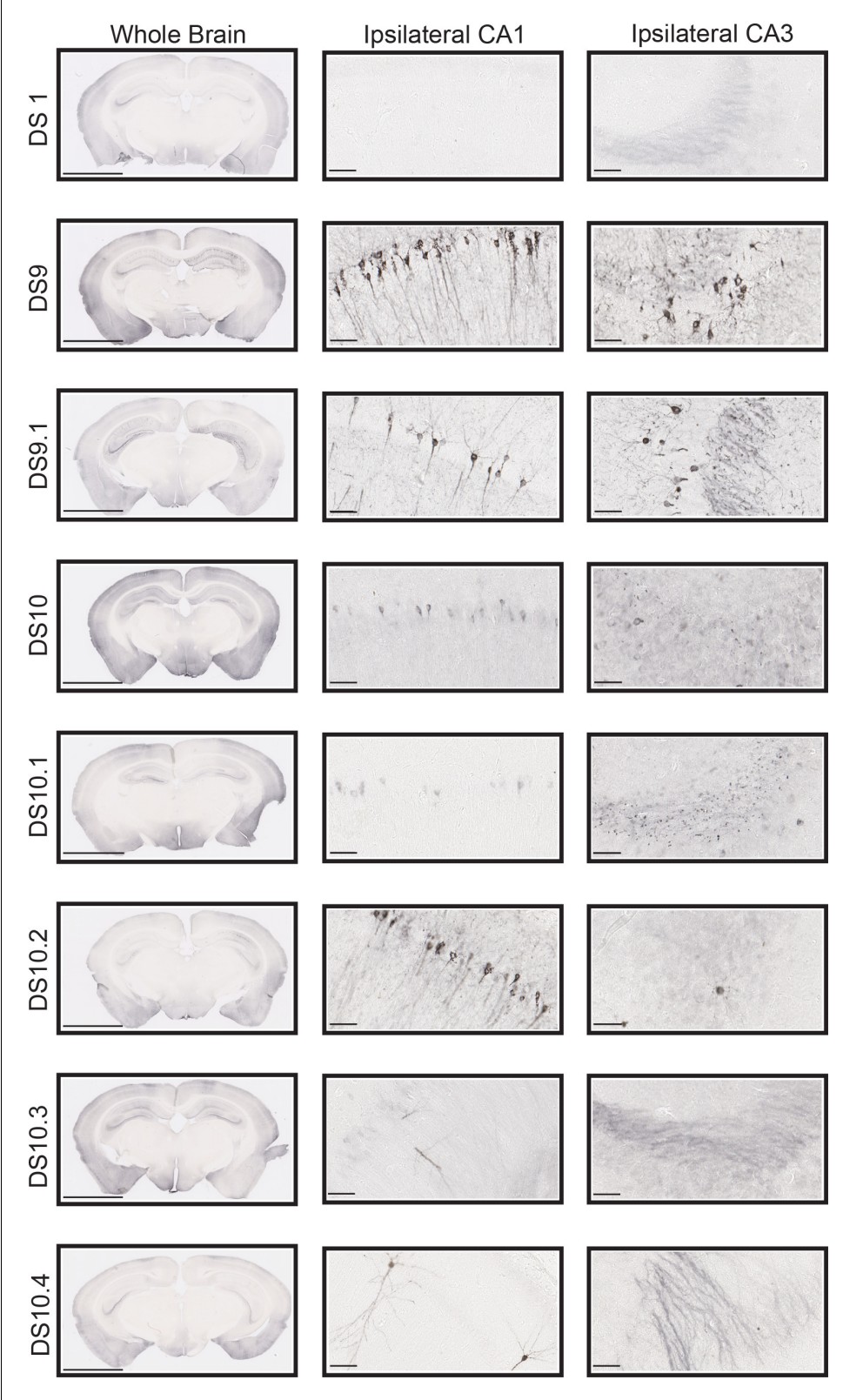

**Figure 3.** Sub-strains trigger unique tau pathology in P301S mice. 10 µg of clarified lysate was injected into the left hippocampi of 3 mo P301S mice, followed by AT8 staining after 4 weeks. Coronal images are oriented with the injection site on the left. DS9 and DS9.1 induced similar pathology in the CA1 and CA3 regions. DS1 induced no pathology. DS10 and DS10.1 induced similar pathology in CA1 and CA3. In both cases, we observed AT8 staining in the cell body throughout CA1 and AT8-positive puncta throughout CA3. DS10.2 induced AT8 signal in both the cell body and along the

*Figure 3 continued on next page*

Figure 3 continued

axons in CA1. There was very little pathology in CA3. DS10.3 induced very little AT8 signal in CA1 and none in CA3. DS10.4 likewise induced little pathology in CA1 region and none in CA3. Each image represents an example from five mice (3 males/2 females or 2 males/3 females per group) treated identically. We noted no differences in induced pathology between males and females. The scale bars represent 200 μm for the whole brain and 50 μm for the CA1 and CA3 regions.

DOI: https://doi.org/10.7554/eLife.37813.009

As we had previously observed (*Sanders et al., 2014*), each clone exhibited a single 'speckled' conformation, indicating that we had most likely propagated a single strain (*Figure 4A*). We also isolated AD $M_s$ by passage through a 100 kD cutoff filter and inoculated DS1 cells, isolating 23 monoclonal lines. All clones had similar morphological characteristics to AD(t) and AD(m) (*Table 5*). To better characterize the tau prions, we performed limited proteolysis on the lysates from AD(t) and AD(m) clones. This revealed identical digestion patterns (*Figure 4B*). We also compared them using sedimentation analysis. Strains derived from AD(m) or AD(t) consisted of both soluble and insoluble protein (*Figure 4C*). Seeding analyses from the cell lines also revealed very similar potencies (*Figure 4D*). We concluded that tau $M_s$ from the AD patient encoded the same single strain as total lysate.

Next we evaluated strains derived from total lysate or monomer from a CBD patient. Upon transduction of the DS1 cells, the total CBD lysate produced two distinct strains: CBD1(t) and CBD2(t). $M_s$ from the CBD brain created two strains identical to those derived from total lysate CBD1(m) and CBD2(m), plus a third strain, CBD3(m) (*Figure 5A*, *Table 6*). CBD $M_s$ was also isolated using 100 kD cutoff filter and transduced into the DS1 line to produce distinct clones. Based on morphology, the proportion of sub-strains was similar to that obtained using gel filtration (*Table 6*). Whether derived from total lysate or monomer, tau from clones CBD1(t) and CBD1(m), or CBD2(t) and CBD2(m) exhibited similar sedimentation properties (*Figure 5B*), proteolysis patterns (*Figure 5C*), and seeding activities (*Figure 5D*). CBD3(m) exhibited a unique morphology, with ordered aggregates (*Figure 5A*), distinct sedimentation (*Figure 5B*), proteolysis pattern (*Figure 5C*), and higher seeding activity (*Figure 5D*).

We concluded that $M_s$ from CBD could be comprised of three independent seed-competent monomers, or that it could occupy a limited conformational ensemble before forming larger assemblies of a particular structure. In the first case, we predicted that $M_s$ from each of the three CBD sub-strains would only produce the parent strain. Conversely, if a conformational ensemble could form three different strains, then $M_s$ from any one of the CBD sub-strains would recreate the entire panel. We isolated monomer from the three CBD sub-strains, re-inoculated DS1, and evaluated the resultant clones by morphology. $M_s$ derived from each sub-strain CBD1 preferentially encoded the parent strain, but also the other two (*Table 7*). These data indicated that $M_s$ derived from CBD

**Table 4.** Analysis of AT8 signal in hippocampi of injected mice.

PS19 mice were inoculated with sub-strains from DS9 and DS10 at 3 months into the hippocampus (n = 5 each). After 8 weeks, coronal sections of hippocampus were analyzed by a blinded reviewer educated on different sections as to the characteristics of each clone. One error occurred in analysis of 40 brains.

| Strain inoculated | Blinded analysis |
| --- | --- |
| DS1 | DS1 were correctly classified. |
| DS9 | DS9 and DS9.1 were indistinguishable. |
| DS9.1 | |
| DS10 | DS10 and DS10.1 were indistinguishable. |
| DS10.1 | |
| DS10.2 | DS10.2 were correctly classified |
| DS10.3 | DS10.3 were correctly classified. |
| DS10.4 | DS10.4 correctly classified 4/5 times, 1/5 incorrectly classified as DS10.3 |

DOI: https://doi.org/10.7554/eLife.37813.010

represented a conformational ensemble (analogous to DS10) that encoded a defined subset of strains. Conversely, $M_s$ derived from the AD patient had a more restricted conformational state that only encoded a single strain.

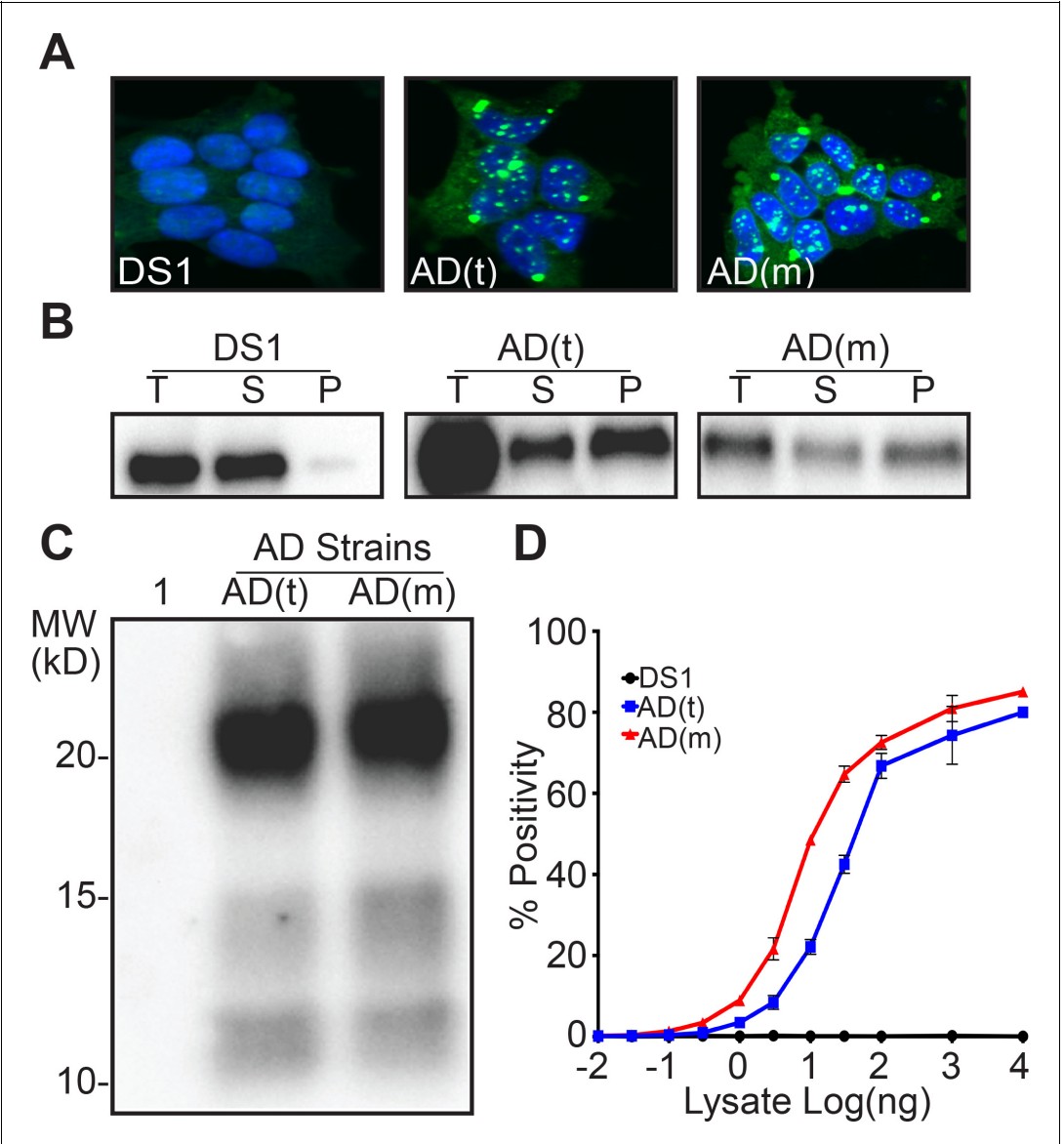

**Figure 4.** $M_s$ derived from an AD patient produces a single strain. (**A**) Clonal cell lines derived from AD-derived total lysate, AD(t), and $M_s$, AD(m) exhibited identical inclusion morphologies. (**B**) RD-YFP derived from AD(t) and AD(m) had similar solubility profiles. RD-YFP in DS1 was only present in the supernatant fraction. Total (T) lysate was resolved into supernatant (S) and pellet (P) fractions by ultracentrifugation. Supernatant to pellet ratio loaded on the gel was 1:1 for all samples. (**C**) Limited proteolysis of RD-YFP aggregates from AD(t) and AD(m) clonal lines produced identical band patterns. Lane 1 represents DS1, which is comprised of RD-YFP monomer that is completely digested. (**D**) Lysates of AD(t) and AD(m) clonal lines had similar seeding profiles. Images are representative of thousands similar cells. Western blots are representative of at least three replicates. Seeding assays represent an individual experiment in which each data point represents a sample analyzed in triplicate. Error bars represent the standard deviation.

DOI: https://doi.org/10.7554/eLife.37813.011

The following source data is available for figure 4:

**Source data 1.** Source data for *Figure 4D*.
DOI: https://doi.org/10.7554/eLife.37813.012

## Discussion

We have previously defined two conformational ensembles of tau: $M_i$, which is inert, and $M_s$, which is competent for self-assembly and seeding (*Mirbaha et al., 2018*). It has been unclear whether monomeric tau encodes strain information, or if this is determined by multimeric assemblies. We now conclude that $M_s$ represents an ensemble of structures, in which each encodes a restricted set of strains. $M_s$ derived from artificially derived strains (DS9 and DS10) replicated the parent strain exclusively (DS9), or a series of sub-strains, one of which closely resembled the parent strain (DS10). These sub-strains each produced unique pathology upon inoculation into the PS19 tauopathy mouse model. Turning to patient-derived samples, we observed that, like total brain lysate, $M_s$ from an AD patient encoded a single strain. Conversely, while total lysate from a CBD patient encoded two strains, $M_s$ derived from CBD lysate encoded these two strains, plus a third. Thus we hypothesize that tau, despite being unstructured by classical biophysical measures, can adopt multiple, stable conformational ensembles in its seed-competent form, $M_s$. These may restrict subsequent assembly to a single fibril conformation (as for $M_s$ derived from DS9 and AD), or may enable formation of a limited number of assembly structures that constitute individual strains (as for $M_s$ derived from DS10 and CBD).

Intriguingly, monomer from DS10 produced five sub-strains. DS10.1 resembled DS10 in every respect, except that monomer from this sub-strain did not recreate the original diversity seen in monomer from DS10. This suggests that in forming DS10.1, $M_s$ adopts a structure very similar to that of DS10 monomer, but with energetic or kinetic barriers that restrict its conformation. It appears that in isolating $M_s$ from DS10 we enabled a new, restricted $M_s$ conformation to emerge and be trapped within the DS10.1 strain. Nonetheless, the overall similarity of DS10 and DS10.1 in terms of inclusion morphology, seeding activity, proteolytic patterns, and neuropathology suggests that the core amyloid conformations are likely to be almost identical.

A major caveat of this work is that we amplified strains using tau RD-YFP containing two mutations (P301L/V337M), not full-length protein. We recognize that our system may bias detection of strains. Also, it remains to be determined (and studies are ongoing) whether RD-YFP captures the critical core structures of fibrils that occur in patients. For example, prior work by Fitzpatrick et.al. indicated that two distinct fibril morphologies (paired helical filaments and straight filaments) could be detected in AD brain (*Fitzpatrick et al., 2017*). These appeared to derive from different configurations of a single monomer template. This is consistent with our prior work, where 2/6 AD patients we analyzed contained two distinct strains, whereas in 4/6 patients, we observed a single strain (*Sanders et al., 2014*). Nonetheless, based on our characterization of strains derived from $M_s$ extracted from the AD and CBD brains, we feel confident in our primary conclusions. In this regard, we were excited to see recent work by Ohhashi et.al, who found that the intrinsically disordered region of Sup35 harbored local compact structure, and also that distinct forms of monomer could give rise to unique Sup35 amyloid conformations (*Ohhashi et al., 2018*).

Based on this work, we predict that related 'families' of strains will be described in patients, linked by structural similarity (*Figure 6*). More specifically, we hypothesize that strain families will utilize distinct combinations of amino acids to form local 'core' structures that are largely preserved, even within a monomer. Similarly, seed competent monomers will adopt a series of structurally related conformations that further assemble into amyloid fibrils. Once a multimeric assembly forms, our observations suggest that it replicates faithfully as a strain. This explains how a single patient

**Table 5.** Sub-strains generated from AD monomer isolated by SEC or cutoff filter.

Tau monomer ($M_s$) from AD brain was purified by immunoprecipitation followed by SEC or passage through a 100kD cutoff filter, prior to inoculation into DS1 cells. Columns indicate the number of clones identified (n) and the percentage this represents of the total (%). A single AD sub-strain was identified regardless of purification method. Classification of cell morphology was performed using blinded analysis.

| $M_s$ | SEC | | 100kD filter | |
|---|---|---|---|---|
| | N | % | N | % |
| AD(m) | 47 | 100 | 23 | 100 |

DOI: https://doi.org/10.7554/eLife.37813.013

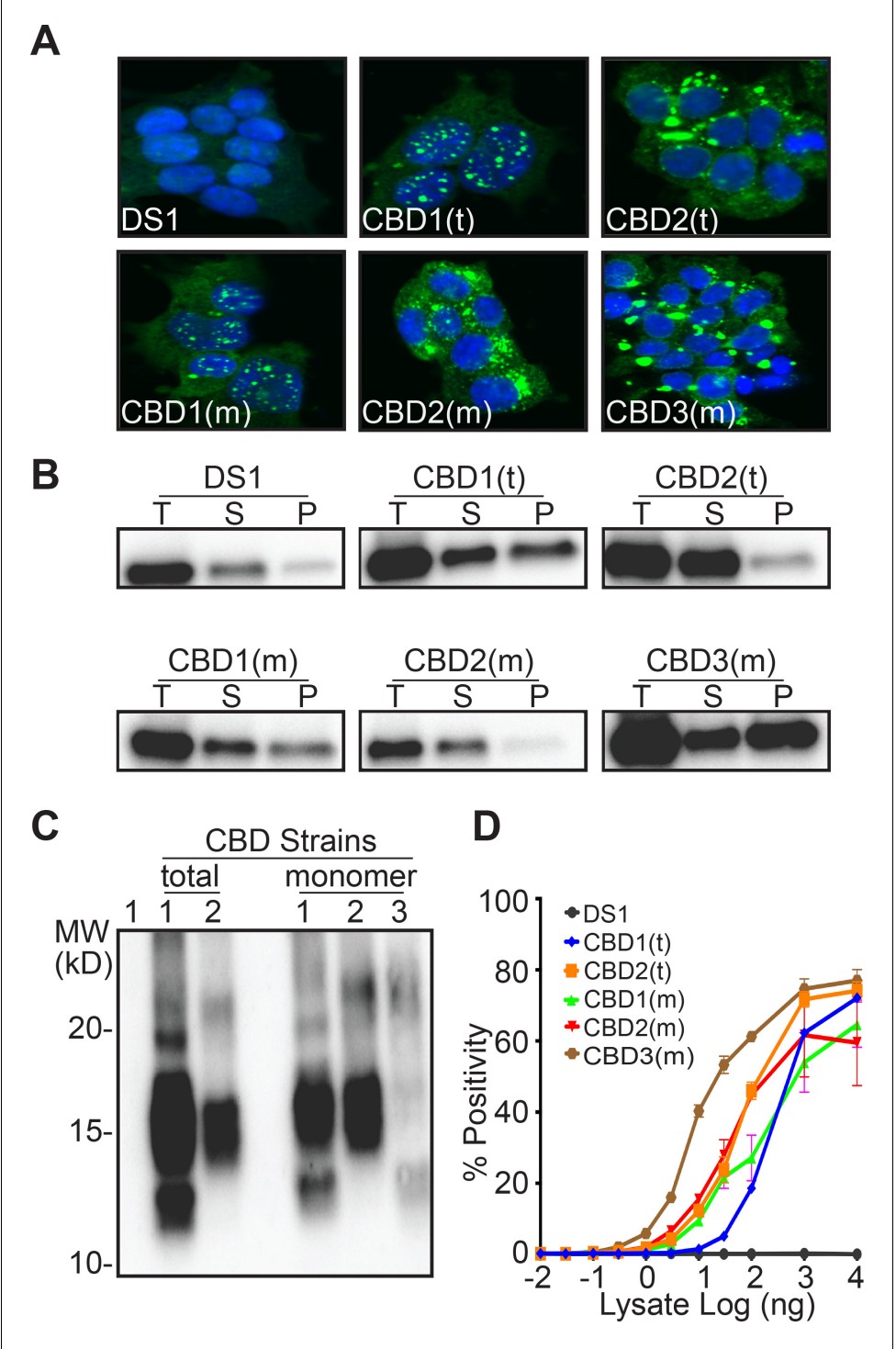

**Figure 5.** $M_s$ derived from a CBD patient produces three distinct strains. (**A**) Clonal lines derived from CBD total lysate had two distinct inclusion patterns, CBD1(t), CBD2(t). Clonal lines derived from CBD $M_s$ had two patterns identical to those from the total lysate, CBD1(m), CBD2(m), and a third, CBD3(m). (**B**) RD-YFP from CBD1(t) and CBD1(m) had mixed solubility. In CBD2(t) and CBD2(m), most RD-YFP was present in the supernatant. CBD3(m) had mixed solubility with most RD-YFP in the insoluble fraction. Total (T) lysate was resolved into supernatant (S) and pellet (P) fractions by ultracentrifugation. Supernatant to pellet ratio loaded on the gel was 1:1 for all samples. Lane 1 represents DS1, which is comprised of RD-YFP monomer that is completely digested. (**C**) RD-YFP aggregates from CBD1(t) and CBD1(m) exhibited similar patterns of proteolysis, with protease-resistant bands around 10–15 kD and a strong band around 15 kD. Proteolysis of RD-YFP from CBD2(t) and CBD2(m) only produced a strong band around 15 kD. CBD3(m) had a unique proteolysis pattern with a band around 10–15 kD. (**D**) Lysates from clones CBD1(t), CBD1(m), CBD2(t) and CBD2

*Figure 5 continued on next page*

*Figure 5 continued*

(m) had similar seeding profiles, while lysate from CBD3(m) had more potent seeding activity. Images are representative of thousands similar cells. Western blots are representative of at least three replicates. Seeding assays represent an individual experiment in which each data point represents a sample analyzed in triplicate. Error bars represent the standard deviation.

DOI: https://doi.org/10.7554/eLife.37813.014

The following source data is available for figure 5:

**Source data 1.** Source data for *Figure 5D*.

DOI: https://doi.org/10.7554/eLife.37813.015

might harbor multiple strains, each potentially derived from a topologically restricted ensemble of tau monomer, $M_s$. By extension, each distinct tauopathy may develop based on emergence of a restricted conformation of $M_s$. This could be due to stochastic or environmental effects, post-translational modifications, or other interactions that facilitate conversion of $M_i$ to a particular ensemble of $M_s$ conformations.

In summary, we present a model to explain how diverse aggregate structures form and give rise to phenotypic diversity in the tauopathies. This predicts a family restricted $M_s$ structural ensembles with varying degrees of relatedness. The original topological restrictions of $M_s$ that give rise to tau prion strains, and the diseases they cause, thus may depend ultimately on initial conformational determinants of tau monomer. Ultimately, an understanding the conformational basis of different tauopathies will facilitate more accurate diagnosis and treatment.

# Materials and methods

**Key resources table**

| Reagent | Designation | Source | Identifiers | Additional information |
|---|---|---|---|---|
| Gene (human) | tau RD (LM)-YFP | PMID: 24857020 | | |
| Cell line (human) | DS1; DS9; DS10 | PMID: 24857020 | | |
| Cell line (human) | Tau RD P301S FRET Biosensor | PMID: 25261551, ATCC | ATCC CRL-3275, RRID:CVCL_DA04 | |
| Cell line (human) | DS9.1; DS10.1; DS10.2; DS10.3,DS10.4, AD(t); AD(m); CBD1(t); CBD2(t); CBD1(m); CBD2(m); CBD3(m) | This paper | | These are cell lines created from tau seeds (total lysate and/or seed-competent monomer) that derived either from clonal cell lines DS9 and DS10, or Alzheimer's or Corticobasal Degeneration disease brain samples that were inoculated into DS1 cell lines, as described in the Materials and Methods, and Results sections. |
| Antibody (rabbit) | TauA (polyclonal against QTAP…KIGSTENL) | This paper | | Antibodies used at dilution indicated in Materials and Methods section (1:1000). |

*Continued on next page*

*Continued*

| Reagent | Designation | Source | Identifiers | Additional information |
|---|---|---|---|---|
| Antibody (rabbit) | Anti-GFP (polyclonal) | Rockland Inc. | Rockland antibodies: 600-401-215, RRID:AB_828167 | Antibodies used at dilution indicated in Materials and Methods section (1:1000) |
| Antibody (mouse) | HJ 9.3 (monoclonal against tau RD) | PMID:24075978, PMID:29566794 | RRID:AB_2721235 | Antibodies used at dilution indicated in Materials and Methods section (1:2000) |
| Antibody (donkey) | ECL Anti-Rabbit | GE Lifesciences | NA9340V, | Antibodies used at dilution indicated in Materials and Methods section (1:2000) |
| Antibody (sheep) | ECL Anti-Mouse | GE Lifesciences | NA931V | Antibodies used at dilution indicated in Materials and Methods section (1:2000) |
| Commercial assay | Amersham ECL Western Blotting reagent | GE Lifesciences | GE Lifesciences: RPN2236 | |
| Commercial assay | 100 kD Spin filter | Corning Spin-X UF | Corning: 431481 | |
| Commercial assay | Agarose beads | Pierce protein A/G Plus | Thermo fisher: 20423 | |
| Software | Graphpad Prism | Graphpad software LLC | | |
| Software | FlowJo | FlowJo LLC | | |
| Other | Lipofectamine 2000 | Thermofisher | Thermo fisher: 11668019 | transfection reagent |

## Cell culture

All cells were grown in Dulbecco's Modified Eagle's medium (Gibco) supplemented with 10% fetal bovine serum (HyClone), 1% penicillin/streptomycin (Gibco), and 1% Glutamax (Gibco). Cells were maintained at 37°C, 5% $CO_2$, in a humidified incubator.

**Table 6.** Sub-strains generated from CBD monomer isolated by SEC or cutoff filter.
$M_s$ from CBD brain was purified by immunoprecipitation followed by SEC or passage through a 100kD cutoff filter, prior to inoculation into DS1 cells. CBD sub-strains, CBD1-3(m), were quantified. Isolation of $M_s$ from CBD brain by SEC or cutoff filter enabled a similar proportion of sub-strains to form. Columns indicate the number of clones identified (n) and the percentage this represents of the total (%). $M_s$ created similar strain patterns regardless of filtration method. Classification of cell morphology was performed using blinded analysis.

| $M_s$ | SEC | | 100kD filter | |
|---|---|---|---|---|
| | N | % | N | % |
| CBD1(m) | 20 | 36 | 4 | 22 |
| CBD2(m) | 18 | 33 | 7 | 39 |
| CBD3(m) | 17 | 31 | 7 | 39 |
| Total | 55 | 100 | 18 | 100 |

DOI: https://doi.org/10.7554/eLife.37813.016

**Table 7.** Quantification of strains derived from CBD-derived sub-strains.

Monomeric RD-YFP derived from each CBD sub-strain was used to inoculate DS1. The resultant clones were then characterized by morphology. $M_s$ from each sub-strain recreated all three, with a preference for the strain of origin. Columns indicate the number of clones identified (n) and the percentage this represents of the total (%). Classification of cell morphology was performed using blinded analysis.

| | Induced clone | | | | | | | |
|---|---|---|---|---|---|---|---|---|
| | CBD1 | | CBD2 | | CBD3 | | Total | |
| Input $M_s$ | n | % | n | % | n | % | n | % |
| CBD1(m) | 30 | 57 | 7 | 15 | 10 | 20 | 47 | 100 |
| CBD2(m) | 9 | 17 | 29 | 62 | 10 | 20 | 48 | 100 |
| CBD3(m) | 13 | 26 | 11 | 23 | 31 | 60 | 55 | 100 |

DOI: https://doi.org/10.7554/eLife.37813.017

## Liposome-mediated transduction of tau seeds

Stable cell lines were plated at a density of 30,000 cells per well in a 96-well plate. After 18 hr, at 60% confluency, cells were transduced with protein seeds. Transduction complexes were made by combining [11.75 µL Opti-MEM (Gibco) +0.75 µL Lipofectamine 2000 (Invitrogen) with cell lysate at a total volume of 25 µL per well. Liposome complexes were incubated at room temperature for 20 min before adding to cells. Cells were incubated with transduction complexes for 48 hr.

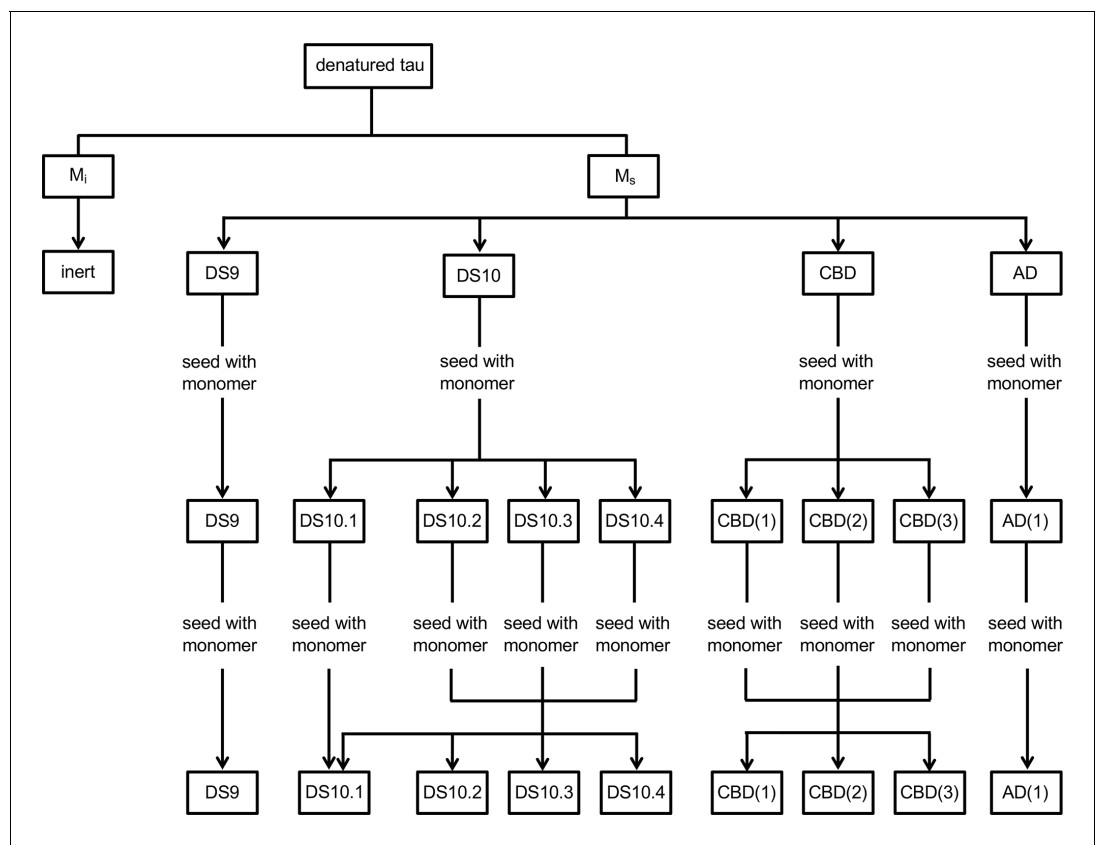

**Figure 6.** Model for families of monomer conformations. We propose a model that discriminates two general conformational ensembles: $M_i$ and $M_s$. $M_i$ is inert, whereas $M_s$ has seeding activity. Within $M_s$, multiple conformations exist that can encode individual or multiple strains. Once an assembly forms, a strain will be faithfully replicated; however, if $M_s$ is isolated from the strain, it can assemble to form a defined set of sub-strains.

DOI: https://doi.org/10.7554/eLife.37813.018

## Monoclonal cell isolation

DS1 cells were treated with clarified lysate and monomer using liposome-mediated transduction of tau seeds. After 48 hr, the cells were harvested and re-suspended in flow buffer (1XHBSS, 1%FBS, 1 mM EDTA). Aggregate-containing cells were identified based on their particularly bright YFP signal using FACS Aria II SORP cell sorter. Cells were sorted individually into a 96-well plate and grown until confluency to derive clonal lines. Lines stably maintaining aggregates were moved to larger plates and amplified for further studies.

## Western blot

Cell pellets were thawed on ice, lysed by triturating in PBS containing 0.05% Triton-X and a cOmplete mini protease inhibitor tablet (Roche), and clarified by 5 min sequential centrifugations at 500 x g and 1000 x g. Total protein concentration of the clarified lysate was measured using Bradford Assay (Bio-Rad). Clarified lysate was mixed with 2X SDS buffer (final SDS concentration 4%) and run on NuPAGE 4–12% Bis-Tris Gel at 150V for ~75 min. The gel was then transferred onto Immobilon P membrane for 1 hr at 20V using a semi-dry transfer apparatus (Bio-Rad). The membrane was then blocked with 5% non-fat dry milk in TBST for 1 hr before primary rabbit anti-tau monoclonal antibody (Tau A, which was raised against QTAP...KIGSTENL) was added at 1:1000 and placed in a shaker overnight at 4°C. The membrane was then washed four times with TBST at 10 min intervals. The membrane was then re-probed with goat anti-rabbit secondary antibody for 1.5 hr at room temperature. The membrane was then washed four times with TBST. Finally, the membrane was exposed to ECL Prime western blot detection kit (GE Lifesciences) for 2 min and imaged with a Syngene digital imager. Images are representative of at least three similar replicates.

## Sonication and size exclusion chromatography

Clarified lysate was sonicated using a Q700 Sonicator (QSonica) at a power of 100–110 watt (Amplitude 50) at 4°C for 1 hr. Samples were then centrifuged at 21000 x g for 10 min and 1 mL of supernatant was loaded into a Superdex 200 Increase 10/300 GL column (GE Healthcare) and eluted in PBS buffer at 4°C. After measuring the protein content of each fraction with a Bradford assay using a plate reader (Tecan M1000), we aliquoted and stored samples at −80°C until further use. Each aliquot was thawed immediately before use. The molecular weight/radius of gyration of proteins in each fraction was estimated by running gel filtration standards (Bio-Rad): Thyroglobulin (bovine) 670 kD/845 nm; γ-globulin (bovine) 158 kD/5.29 nm; ovalbumin (chicken) 44 kD/3.05 nm; myoglobin (horse) 17 kD/2.04 nm; and vitamin $B_{12}$ 1.35 kD/0.85 nm.

## Size cutoff filtration

Monomer fraction from SEC was passed through a 100 kDa MWCO filter (Corning) as instructed by the manufacturer (centrifuged at 15,000 x g for 15 mins at 4°C). Filtered material was immediately collected and protein concentration was determined. The filtrate was aliquoted and frozen in −80°C.

## Monomer isolation

Clarified lysate was ultra-centrifuged at 186,000 x g for 1 hr, followed by washing the pellet with 1 mL PBS. The sample is ultra-centrifuged again at 186,000 x g for 30 mins and the supernatant aspirated. The pellet is re-suspended in 200 μL PBS. The pellet is then sonicated using a Q700 Sonicator (QSonica) at a power of 100–110 watt (Amplitude 50) at 4°C for 1 hr. Samples were then centrifuged at 21,000 x g for 10 min and 1 mL of supernatant was loaded into a Superdex 200 Increase 10/300 GL column (GE Healthcare) and eluted in PBS buffer at 4°C. The eluate was quantified using BSA assay. The monomer fraction showed no seeding in the absence of lipofectamine but showed seeding in its presence (data not shown).

## FRET flow cytometry

Tau P301S Biosensor cells (RRID:CVCL_DA04) were harvested with 0.05% trypsin and fixed in 2% paraformaldehyde (Electron Microscopy Services) for 10 min, then resuspended in flow cytometry buffer. The MACSQuant VYB (Miltenyi) was used to perform FRET flow cytometry. To measure CFP and FRET, cells were excited with a 405 nm laser, and fluorescence was captured with 405/50 nm and 525/50 nm filters, respectively. To measure YFP, cells were excited with a 488 nm laser and

fluorescence was detected with a 525/50 nm filter. To quantify FRET, we used a gating strategy similar to that previously described (*Holmes et al., 2014*). Percent FRET positivity was used to determine the fraction of cells containing aggregates. For each experiment,~10,000 cells were analyzed in triplicate. Analysis was performed using FlowJo v10 software (Treestar). Error bars represent the standard deviation.

### Protease digestion

Pronase (Roche) was diluted in PBS to a final concentration of 1 mg/mL and single-use aliquots were stored at −80°C. Clarified cell lysate was prepared as previously described and protein concentrations were normalized to 4 µg/µL, unless otherwise noted. 40 µg (10 µL) of cell lysate was added to 10 µL of pronase at a concentration of 60 µg/mL (diluted in PBS) for a final volume of 20 µL and a final pronase concentration of 30 µg/mL. Cell lysates were digested at 37°C for 90 min. Reactions were quenched by addition of 20 µL of 2x sample buffer (final SDS concentration of 4%) and boiling for 5 min. 15 µL of each sample was loaded onto a 10% Bis-Tris NuPAGE gel (Novex by Life Technologies) and run at 150 V for 65 min. Protein was transferred to Immobilon P (Millipore) using a semi-dry transfer apparatus (Bio-Rad) and membranes were probed for tau RD as described above.

### Sedimentation analysis

Clarified cell lysate was prepared as described previously. 10% of the lysate was set aside as the total fraction; the rest was centrifuged at 186,000 x g for 1 hr. The supernatant was placed aside and the pellet was washed with 1 mL PBS prior to ultracentrifugation at 186,000 x g for 30 min. The supernatant/wash from this step was aspirated and the pellet was re-suspended by boiling in RIPA buffer with 4% SDS and 100 mM DTT. Bradford assay (Bio-Rad) with BSA standard curve was used to normalize all protein concentrations. 1 µg of total protein was loaded per well on a 4–12% Bis-Tris gel (Invitrogen). For all samples 1:1 ratio of supernatant to pellet was used. Gels were analyzed by western blot.

### Confocal microscopy

96-well plates (Costar 3603) were coated with 1X Poly D-Lysine (PDL), incubated overnight at 37°C. Plates were then washed with PBS and cells were plated and grown in DMEM media for 24 hr. Media was then removed and replaced with 4% PFA for 10 min. PFA was removed and washed 2x with PBS followed by staining with DAPI for 10 min in 0.05% Triton-X. Cells were washed and stored in PBS. The plate was imaged with In Cell Analyzer 6000 at 40x resolution with the assistance of the UTSW high-throughput screening core facility. Images were coded and a blinded counter scored aggregate morphology, blinded to conditions.

### Cell lysate production for animal inoculation experiments

The cell lines were grown in 10 cm dishes until 80% confluency. The cells were then washed, trypsinized, resuspended in media and centrifuged at 1000 x g. Cell pellets were washed with PBS. The pellet was then stored at −80°C. Prior to analysis, pellets were thawed on ice and re-suspended in 1x PBS with cOmplete protease inhibitors (Roche) and sonicated using an Omni-Ruptor 250 probe sonicator at 40% power for 35 × 3 s cycles. The probe sonicator was washed with 10% bleach, 100% ethanol and ddH$_2$O between cell lines. Strains were subsequently centrifuged at 1000 x g, normalized to 7 µg/µL by Bradford assay (Bio-Rad) and stored in aliquots at −80°C.

### Animal maintenance

We obtained transgenic mice that express 4R1N P301S human tau under the murine prion promoter (*Yoshiyama et al., 2007*) from Jackson Laboratory, and maintained them on a B6C3 background. Transgenic mice and wild-type littermates were housed under a 12 hr light/dark cycle, and were provided food and water *ad libitum*. All experiments involving animals were approved by the University of Texas Southwestern Medical Center institutional animal care and use committee.

### Inoculation of mouse brain

P301S mice were anesthetized with isoflurane and kept at 37°C throughout the inoculation. Mice were injected with separate 10 µL gas-tight Hamilton syringes for each strain at a rate of 0.2 µL/min.

Animals were inoculated with 10 μg (1.428 μL) of cell lysate in the left hippocampus (from bregma: −2.5 mm posterior, −2 mm lateral, −1.8 mm ventral). Animals were then allowed to recover and monitored for 40 days prior to tissue collection.

### Animal tissue collection

P301S or WT mice were anesthetized with isoflurane and perfused with chilled PBS + 0.03% heparin. Brains were post-fixed in 4% paraformaldehyde overnight at 4°C and then placed in 30% sucrose in PBS until further use.

### Histology

Brains were sectioned at 50 μm using a freezing microtome. Slices were first blocked for 1 hr with 5% non-fat dry milk in TBS with 0.25% Triton X-100 (blocking buffer). For DAB staining, brain slices were incubated with biotinylated AT8 antibody (1:500, Thermo Scientific) overnight in blocking buffer at 4°C. Slices were subsequently incubated with the VECTASTAIN Elite ABC Kit (Vector Labs) in TBS prepared according to the manufacturer's protocol for 30 min, followed by DAB development using the DAB Peroxidase Substrate Kit with the optional nickel addition (Vector Labs). Slices were imaged using the Olympus Nanozoomer 2.0-HT (Hamamatsu).

## Acknowledgements

We thank Sushobhna Bhatra, Dana Dodd, Lukasz Joachimiak, Hilda Mirbaha, Victor Manon, William Russ, Barbara Stopschinski, Jaime Vaquer-Alicea, and Amy Zwierzchowski-Zarate for critiques. This work was supported by grants from the Tau consortium and NIH grants awarded to 1R01NS071835 (MID), R01NS089932 (MID). We appreciate the help of High Throughput Screening Core (HTS) administered by Bruce Posner, PhD. Human tissue samples were provided by the Neurodegenerative Disease Brain Bank at the University of California, San Francisco.

## Additional information

### Funding

| Funder | Grant reference number | Author |
| --- | --- | --- |
| National Institutes of Health | 1R01NS071835 | Marc I Diamond |
| National Institutes of Health | R01NS089932 | Marc I Diamond |
| Rainwater Charitable Foundation | Tau Consortium | Marc I Diamond |

The funders had no role in study design, data collection and interpretation, or the decision to submit the work for publication.

### Author contributions

Apurwa M Sharma, Conceptualization, Supervision, Writing—original draft, Project administration, Writing—review and editing; Talitha L Thomas, Conceptualization, Data curation, Formal analysis, Investigation, Methodology, Writing—original draft, Writing—review and editing; DaNae R Woodard, Data curation, Formal analysis, Investigation, Methodology; Omar M Kashmer, Formal analysis, Investigation, Methodology, Writing—original draft; Marc I Diamond, Formal analysis, Investigation, Methodology, Writing, Funding acquisition

### Author ORCIDs

Apurwa M Sharma  http://orcid.org/0000-0002-6526-3747
Marc I Diamond  http://orcid.org/0000-0002-8085-7770

### Ethics

Animal experimentation: This study was performed in strict accordance with the recommendations in the Guide for the Care and Use of Laboratory Animals of the National Institutes of Health. All of

the animals were handled according to approved institutional animal care and use committee (IACUC) protocols (#2015-100975) of the University of Texas Southwestern Medical Center.

## Decision letter and Author response

Decision letter https://doi.org/10.7554/eLife.37813.022
Author response https://doi.org/10.7554/eLife.37813.023

## Additional files

### Supplementary files

• Transparent reporting form
DOI: https://doi.org/10.7554/eLife.37813.019

### Data availability

All data generated or analyzed during this study are included in the manuscript and supporting files. Source data have been provided for Figures 1, 2, 4 and 5.

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
