## [Decision Letter]

Thank you for submitting your article "Tau monomer encodes strains" for consideration by *eLife*. Your article has been reviewed by three peer reviewers, and the evaluation has been overseen by a Reviewing Editor and Anna Akhmanova as the Senior Editor. The following individual involved in review of your submission has agreed to reveal his identity: Jeffery W Kelly (Reviewer #2).

Reviewers discussed the reviews with one another and the Reviewing Editor drafted this decision to help you prepare a revised submission.

Summary:

This report provides support for the concept that tau monomers contain information that will seed specific types of tau aggregates, designated in this manuscript as "tau strains". The report is an extension of prior report showing that monomer liberated from fibrils can retain structural features that promote seeding capacities, leading to the conclusion that tau monomers are either seeding competent (M_s_) or seeding inert (M_i_). Importantly the Tau M_s_ retains additional structural information that produces distinct cellular tau assemblies. This is potentially a revolutionary idea that could have significant impact on understanding the structural basis for tauopathies as well as other diseases associated with protein aggregation.

Essential revisions:

1) While authors use several approaches to minimize potential contamination, use of the cell derived material poses some complications. For example, if potential "soluble" oligomers are proteolytically processed within the cell, filtration using 100kDa cut-off filters may not remove any "processed" oligomers. While one could attempt to detect such "oligomers" but such effort is challenged if the oligomers are present at very low levels. One suggestion would be to see if depletion of YFP-containing "monomers" significantly reduce the "seeding" capacity of the monomers from the clones. While this is not perfect, at least the authors could rule out any spurious oligomers containing the only the "tau RD" domain.

2) Use of "strain" in the context of this report indicate direct structure determinant. The outcome in this report is completely based on the gross cellular morphology of "aggregates" and neuroanatomical distribution of tau pathology. Other than protease sensitivity assay of cell derived material, there is no demonstration that strains are structurally distinct. For example, do the cell derived monomers directly seed in vitro aggregation of tau fibrils with different structures/morphologies?

3) At various places in the manuscript that they consider tau as a prion. Tau should not be considered as a prion yet, because tau still lacks certain properties that classifies prions. Most importantly, no epidemiological data exit that demonstrate the inter-individual transmissibility of tau aggregates in humans. As long as tau does not meet all criteria classifying prions, it should not be designated as a prion.

---

## [Author Response]

Essential revisions:1) While authors use several approaches to minimize potential contamination, use of the cell derived material poses some complications. For example, if potential "soluble" oligomers are proteolytically processed within the cell, filtration using 100kDa cut-off filters may not remove any "processed" oligomers. While one could attempt to detect such "oligomers" but such effort is challenged if the oligomers are present at very low levels. One suggestion would be to see if depletion of YFP-containing "monomers" significantly reduce the "seeding" capacity of the monomers from the clones. While this is not perfect, at least the authors could rule out any spurious oligomers containing the only the "tau RD" domain.

The reviewer raises a relevant question of whether in DS cell lysates a tau RD cleavage product lacking the bulky GFP could encompass the tau RD fragment, which at ~14kDa could form cryptic multimers that would pass through the 100kDa filter and otherwise be invisible to our assays. To address this concern, we precipitated seeding activity from DS10.2 cell lysates using HJ9.3 (for reasons we have found it impossible to IP seeds using anti-GFP antibody). HJ9.3 immunoprecipitated the entirety of seeding activity (Author response image 1).

**Author response image 1. respfig1:** Immunoprecipitation of seeding activity from DS10.2. Lysates from DS10.2 cells expressing tau RD-YFP in an aggregated form were exposed to HJ9.3 antibody, which recognizes tau aa306-321, followed by immunoprecipitation. Total, supernatant, and pellet fractions were transduced via Lipofectamine into tau RD-CFP/YFP FRET biosensor cells, and the percentage of cells with aggregates was quantified by flow cytometry. After exposure to anti-tau antibody, no seeding activity remained in the supernatant, and all had been removed to the pellet fraction.

To test for the possibility that an RD fragment lacking the GFP epitope was accounting for the strain activity, we probed the western blots with HJ9.3, and anti-GFP antibody. We detected two bands with each antibody (Author response image 2). The dominant band, at ~45kDa is consistent with full-length RD-YFP. A smaller band at ~35kDa was detected by both HJ9.3 and anti-GFP antibodies (Author response image 2). We did not observe bands whose reactivity is consistent with a cleavage product of RD alone (which would be closer to ~14kDa). In parallel, we carried out studies of tau monomer from AD brain, which, like RDYFP, carries strain information. We observed no evidence of tau RD fragments.

**Author response image 2. respfig2:** Tau monomer from DS10.2 is predominantly full-length. Diagram indicates tau RD-YFP and the epitope of HJ9.3, which was used for initial

immunoprecipitation (IP). Following IP, SDS-PAGE was performed, followed by western blot using secondary antibodies of either HJ9.3 (left) or anti-GFP (right). HEK cell lysate was negative control. AD brain lysate was also analyzed. Tau RD-YFP at ~45kDa is the predominant band, with a secondary band (which is only occasionally observed, but recorded here) at ~35kDa that contains the GFP epitope and the HJ9.3 epitope. Note that tau monomer derived from AD brain (which also encodes strains) has no obvious truncation products.</Author response image 2 title/legend>

In summary, while we cannot exclude the possibility that a small number of invisible RD oligomers of tau are encoding strains within the cell lines, our biochemical studies speak to the dominant form of tau monomer as intact RD-YFP, with a minor product that is a small fraction the sample (and, indeed, is not consistently visible when we perform repeated studies) that contains both RD and YFP epitopes. Finally, with regard to the central point of our manuscript, we have observed no fragments of full-length tau that can account for strain-encoding seeding activity as would be predicted by cryptic monomers, and so we stand by our central conclusions.

2) Use of "strain" in the context of this report indicate direct structure determinant. The outcome in this report is completely based on the gross cellular morphology of "aggregates" and neuroanatomical distribution of tau pathology. Other than protease sensitivity assay of cell derived material, there is no demonstration that strains are structurally distinct. For example, do the cell derived monomers directly seed in vitro aggregation of tau fibrils with different structures/morphologies?

The reviewer brings up a valid point that has bedeviled the field of prion research, although we strongly disagree with the statement that we have provided “no demonstration” that strains are structurally distinct – we have used inclusion morphology, seeding activity, biochemical characteristics, inoculation in vivo, and limited proteolysis. Together these represent a fairly comprehensive, if indirect, assessment of a prion’s conformation, or strain identity. In fact, we would like to emphasize that what we have done exceeds what is typically performed in PrP research to discriminate strains. It is currently impossible in our lab (and in others, as far as we know) to faithfully amplify a single tau prion strain in vitro. Thus we are left with the above mentioned “indirect” structural probes, the best of which appears to be limited proteolysis. While we are working on this problem, so far we have failed to develop a method to directly determine the conformation of tau prion strains. Within the limits of our experimental expertise we feel safe in concluding we are in fact generating distinct strains. It is worth mentioning that in our analyses of over two dozen distinct strains, we have never observed that strains with different cell morphological patterns (which we used initially to select distinct strains) harbor the same strain. The reverse is not true. Cells that have similar inclusion morphological can in fact harbor distinct strains. So if anything, by using cell morphology as a probe we are *underestimating* the prevalence of distinct strains.

3) At various places in the manuscript that they consider tau as a prion. Tau should not be considered as a prion yet, because tau still lacks certain properties that classifies prions. Most importantly, no epidemiological data exit that demonstrate the inter-individual transmissibility of tau aggregates in humans. As long as tau does not meet all criteria classifying prions, it should not be designated as a prion.

The reviewer brings up a valid and common criticism of our use of the term “prion.” However, even Stanley Prusiner, who invented the term, now refers to tau as a “prion” in his work. In our 2014 publication in Neuron we carried out a series of experiments that we feel are definitive in establishing tau as a bona fide, infectious prion. We created infectious tau by in vitro fibrillization reactions, inoculated cell lines, isolated distinct strains that stably propagated transmissible tau species in vitro, and created transmissible pathology across three generations of mouse models through brain inoculation. In our opinion, the lack of observed human transmissibility “in the wild,” is not a reason to use a different term for tau prions, as in every respect in a laboratory environment they meet the criteria that have been used to study and propagate PrP prions for decades.

We would also point out that without extraordinary transmission events such as extreme agricultural practices (e.g. feeding sheep offal to cows), cannibalism and inoculation through tissue transplantation or brain surgery, even human PrP disease would not spontaneously transmit between individuals. Now that commonsense practices are in place there is essentially no human-human transmission of PrP prion disease. Thankfully, there are no known cases of tau transmission through tissue transplantation or brain surgery, although the absence of this type of inter-person transfer (which would presumably satisfy the reviewer that tau is a prion) does not mean it hasn’t happened, or couldn’t happen. Certainly our prior transmission of tau pathology in mice suggests transmission between individuals would be feasible under unfortunate circumstances.

In summary, we respectfully request that we keep this term, as for all practical purposes, we maintain that tau meets the biochemical, biological, and experimental criteria of a prion.